# A Bus Network Design Model under Demand Variation: A Case Study of the Management of Rome's Bus Network

Andrea Gemma [1], Ernesto Cipriani [1], Umberto Crisalli [2], Livia Mannini [1,*] and Marco Petrelli [1]

1   Department of Civil, Computer Science and Aeronautical Technologies Engineering, Roma Tre University, 00146 Rome, Italy; andrea.gemma@uniroma3.it (A.G.); ernesto.cipriani@uniroma3.it (E.C.); marco.petrelli@uniroma3.it (M.P.)
2   Department of Enterprise Engineering, Tor Vergata University of Rome, 00133 Rome, Italy; crisalli@ing.uniroma2.it
*   Correspondence: livia.mannini@uniroma3.it

**Abstract:** This paper proposed a methodology to design bus transit networks that can be consistently adjusted according to demand variations both in level and distribution. The methodology aims to support the activities of service providers in optimizing the service capacity of the bus network according to a system-wide analysis. It stems from the changes imposed by the COVID-19 pandemic. Such an experience has imposed a rethinking of the methodology used for the optimal design of robust transit network services that are easy-to-adapt to demand variations without redesigning the whole network every time. Starting from an existing model, this design methodology is articulated in two parts: the first part for solving the problem with the maximum level of transit demand, aiming at giving an upper bound to the solution, and the second part, where the network is optimized for other specific transit demands. This method has been applied to a real context in the city of Rome, considering two levels of demand taken from COVID-19 experiences. They are characterized by the application of different policies regarding different timings for shopping and schools' openings as well as by policies on smart working. The results show the effectiveness of the proposed methodology to design robust transit networks suited to comply with large demand variations. Moreover, the procedure is suitable and easy to implement, in order to adapt quickly to changes in demand without having to modify line routes, but adapting them in an optimal way, even when dealing with realistic-sized transit networks.

**Keywords:** bus network design; demand variation; COVID-19 pandemic

## 1. Introduction

In recent years, public transport (PT) has faced a great challenge as a result of the exceptional decrease in transit ridership due to the pandemic. Specifically, transit demand is subject to large and sudden structural variations due to post-pandemic user behaviour, for which the development of new network planning requirements is needed. These structural changes in demand that can occur repeatedly and close in time (from month to month) can be accompanied by designing bus networks that result from a quick reallocation of necessarily constant resources.

The aim of this paper is to provide a methodology to design bus transit networks to be consistently adjusted according to demand variations both in level and distribution. Therefore, our methodology aims to support the activities of service providers in optimizing the service capacity of the network according to a system-wide analysis. Currently, these decisions are seldom based on such an approach, and this represents an urgent requirement for service providers, as stated by Gkiotsalitis and Cats [1] during pandemic.

The proposed method is suitable to manage adaptations in transit services consistently with demand changes in both level and distribution due to variations induced by different timings of shopping and schools' openings, as well as by work-at-home policies (smart

working). This proposal was born from the habit changes in our lives imposed by the COVID-19 pandemic, which resulted in the rethinking of the peak hour concept and needs an easy-to-apply methodology from the view of planners for the optimal design of transit network services that are easy to rapidly adapt to demand variations without redesigning the whole network every time. The method allows to make the supply system more effective from the users' point of view and more efficient from the operators' point of view. Therefore, public transport services are more and more capable of satisfying the requests of citizens and competitive with private transport, thus, the urban mobility system can become more sustainable. Specifically, the application of the proposed method leads to a supply system which is both environmentally and economically more sustainable.

This method has been applied to a real context in the city of Rome, considering two levels of demand taken from the COVID-19 experience, coming from government restrictions to shopping times, school timetables, and smart working for non-essential workers.

The paper is structured as follows. Section 2 reports the state of the art. Section 3 describes the proposed methodology. Section 4 presents the results of the case study carried out in the city of Rome. Finally, Section 5 summarizes the conclusions and future developments of this research.

## 2. Literature Review

The literature on transit network design (TND) applications is wide and well-consolidated, however, very few papers treat the TND problem as related to demand variation, except for some very recent papers on the effects of the COVID-19 pandemic on public transport. In this last field, Gkiotsalitis and Cats [1] provided a literature review on the impacts of the pandemic on public transport, identifying planning measures and intervention measures to support public transport operators. In Tirachini and Cats [2], the authors reported the state of the art up to June 2020 of public transportation actions adopted by governments and agencies across the world. The TND problem deals with the identification of the optimal network configuration in terms of routes and frequencies in order to minimize the objective function, which is usually represented by both passengers' and operators' costs.

TND represents a challenging non-convex issue, as described in [3,4]. It can be addressed as an optimization problem that is non-linear, involving a combination of continuous and discrete constraints and variables. Many authors have studied and addressed the issue of TND. Among the first to face and solve the problem, we refer to [5], who proposed a TND procedure in which, after the generation of a large set of feasible routes connecting every node to all others, the system creates subsets of routes solving a Set Covering Problem. Carrese and Gori [6] developed a bus transit network design (BTND) procedure that generates a hierarchical transit system articulated in express, main and feeder lines. Lee and Vuhic [7] presented an iterative procedure which forms a set of routes consisting of OD shortest paths, followed by the elimination of the less efficient ones. Contrarily, Wang and Lo [8] transformed the TND problem into a single-level model, and so solved it in a mixed-integer linear program (MILP). The procedure involves two steps: the first step linearizes the constraints and the second step linearizes the objective function.

The TND optimization problem has often been tackled using various methods such as Genetic Algorithms (GA), Tabu Search and Simulated Annealing. Duran-Micco et al. [9] provided a comprehensive review of the TND literature, focusing on studies published within recent years. Cipriani et al. [10] described a procedure to solve the BTND based on two stages; the first consists of a heuristic algorithm that provides a set of feasible routes, while the second uses a GA to find the optimal network regarding frequencies and routes. Ciaffi et al. [11] presented a three-step procedure for solving the BTND for a multimodal transit system: the first step identifies the zones to be served, the second step generates feasible routes by means of a heuristic algorithm and, finally, the third step applies a GA for choosing the set of routes and their frequencies. Nayeem et al. [12] developed a population-based model for TND based on GA optimization, maximizing the number of satisfied

passengers and minimizing the transfers and the travel time. Bourbonnais et al. [13] performed optimizations with GA using accurate data on the road network and reliable data on public transport demand, thus leading to a more efficient network using a comparable fleet size and the same parameters. Arbex et al. [14] studied an Alternating Objective Genetic Algorithm (AOGA) for solving a transit network design and frequency setting problem (TNDFSP), where the objective to be investigated was cyclically alternating over generations. Micco et al. [15] considered in a TNDFSP a combined maximum frequency between all lines using a subset of chosen links, addressing any crowding problems. Other methods involved ant colony optimization, as in [16], and Bee Colony Optimization (BCO), as in [17], which solved the TND problem by means of Swarm Intelligence (SI) based on the BCO's metaheuristics. Later, Nikolić et al. [18] developed a procedure based on the BCO's metaheuristics for TND that simultaneously determines the links to be assembled in the routes and the bus frequencies. Szeto et al. [19] and, later Liu et al. [20], formulated an objective function concerning the number of passengers without direct service and evaluating the total the total travel time for the passengers served directly. Pinelli et al. [21] proposed a data-driven TND deriving patterns from mobile phone location data; the latter are merged to identify the candidate routes. Additionally, Bertsimas et al. [22], that presented a data-driven TND applied to a real network in Boston. An and Lo [23] formulated a stochastic program, developed in two stages. In the first, the transit line alignments and frequencies were identified; in the second stage flexible services were determined to capture the cost of the demand overflow. Finally, the solution algorithm applied to the networks combined the gradient method and neighbourhood search. Calabrò et al. [24] proposed a flexible transit design that makes optimal use of fixed-route and dial-a-ride transit, depending on the demand observed in a specific urban agglomeration and at different times of day. Huang et al. [25] investigated a multimodal TND, introducing a hub-and-spoke network framework, thus considering the rail system as the core. The main bus services were designed on the basis of a heuristic line generation algorithm; in order to integrate the rail and the feeder bus design, a travelling salesman problem was solved. Finally, the frequencies were identified with a bi-level programming model, applying the artificial bee colony algorithm. Some authors have included different objectives in addition to the classic costs. Cancela et al. [26] presented TND from a mathematical programming point of view. Specifically, the definition problem of the bus line's number and routes is considered, as well as its frequency. The authors applied mixed integer linear programming (MILP), including the waiting time and the existence of multiple lines in the users' behaviour. Tong et al. [27] designed a TND to maximize system-wide transportation accessibility. Pternea et al. [28] developed a sustainable TND that incorporated sustainable design objectives, considering electric vehicles and introducing a direct route design approach. Cheng et al. [29] studied the impact of decreasing transit performances on emissions in a single-technology transit system and extended their analysis to a transit system with a hierarchical structure, considering an elastic demand. The authors presented a base for designing urban transit systems while reducing GHG emissions and social costs. Feng et al. [30] proposed a GA to solve a TND problem, strengthening the effect of the transfer time on the total transit trip time. Farahani et al. [31] provided a review of a TND problem, specifically focusing on the solution methods for the urban transportation network design problem (UTNDP), which takes into account both the Road Network Design Problem (RNDP) and the Public Transit Network Design Problem (PTNDP). Kennedy and Eberhart [32] introduced a concept for the optimization of nonlinear functions using particle swarm (PSO) methodology. In the particle swarm optimization (PSO) algorithm, the optimum solution is searched for in a multidimensional space traversed by the particles. Each of the latter is a potential solution and it is influenced by the experiences of its neighbours. Sengupta et al. [33] provided a review of the PSO algorithm's applications. The authors highlight that the PSO can be used on any objective function. Miandoabchi et al. [34] dealt with TND by also considering the car and bus flow interaction as being expressed as a multi-objective optimization model. The problem was solved by a hybrid GA and a hybrid clonal selection algorithm. Subse-

quently, Miandoabchi et al. [35] fused the road and the bus network design decisions, and investigated, in addition to the hybrid GA, the PSO and harmony search into which the simulated annealing was incorporated. Kechagiopoulos and Beligiannis [36] implemented and applied a PSO-based algorithm to the Urban Transit Routing Problem (UTRP). Their results are compared to Mandl's benchmark problem. Hassannayebi et al. [37] proposed a PSO algorithm to minimize the passenger waiting time. Zhong et al. [38] proposed a PSO method for identifying rapid transit bus routes optimized to maximize the number of passengers. Buba and Lee [39] solved the urban TND problem by means of a hybrid differential evolution with particle swarm optimization (DE-PSO), simultaneously optimizing the routes and frequencies. Lopez et al. [40] presented a PSO algorithm to optimize a mass transit system's electrical infrastructure, where the design of the PSO parameters led to good element-speed models. Jha et al. [41] studied a multi-objective TNDFSP, which was solved in two stages. The first concerns the identification of a set of routes based on an initial route set generation (IRSG) procedure together with a GA. The second stage deals with the assignment of the frequency of routes solved as a multi-objective particle swarm optimization (MMOPSO). Park et al. [42] presented a multi-objective solution for addressing the TNDP in the presence of fluctuating demand, taking into account considerations related to transit equity. Amiripour et al. [43] developed a hybrid method to optimize the design of a bus network, considering the seasonal variation of passenger demand. Later, Amiripour et al. [44] provided a GA procedure to design a bus network with seasonal variations of demand. These papers represent some of the first attempts to solve the BTND problem with variations of passenger demands and the developed method was a hybrid solution procedure, called HBRD-I; first creating different networks, applying GAs for different demand scenarios, and then searching for a hybrid solution by selecting routes using an overlapping score and satisfied demand level in every scenario. Therefore, in both the last two papers, there is a clear indication of the need to take explicitly into account the demand variations to provide a convenient service across all demand conditions, where a convenient service is one able to maximize the given demand.

Starting from these considerations, the aim of this paper is to provide a procedure for transit network design that can consider demand variations. In the COVID-19 pandemic era the variation of demand is not only due to seasonality but also to planning actions and the impacts of the crisis. This methodology uses a GA algorithm, although it could use a PSO one; whichever algorithm you want to use will be efficient, and they represent a tool that does not impact the overall proposed methodology.

## 3. Methodology

Our proposed methodology is articulated in two parts: the first part, called primary BTND, is aimed at solving the problem for the maximum level of transit demand, aiming at giving an upper bound to the solution, and the second part, called secondary BTND, is where the network is optimized for other specific transit demands (e.g., in levels, usually reduced, and distributions). The functional architecture of this methodology is shown in Figure 1. Specifically, the proposed BTND methodology, both for the primary and the secondary part, is made of two phases: the first allows us to generate a set of feasibility routes; the latter allows us to define the optimal set of bus routes, including their frequencies. The proposed methodology implies an iterative approach in which changes in the "line pool" to feasible routes allows us to "fish" for an optimal subset of routes and relative frequencies able to satisfy the demand variations in both level and distribution. In particular, for primary BTND, the collection of possible routes is generated by the Heuristic Route Generation Algorithm (HRGA) while, for secondary BTND, the set of feasible routes (Phase 1) is the optimal solution proposed by the primary one.

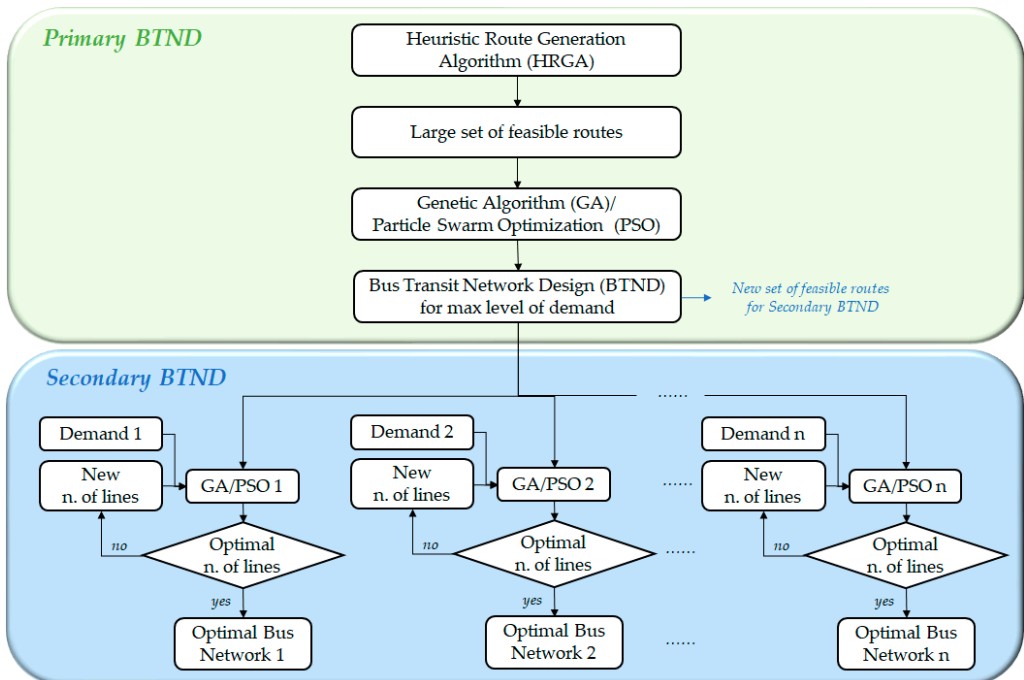

**Figure 1.** Functional architecture.

In the primary part, the set of feasibility routes (Phase 1) is defined by using an heuristic approach, generating a mix of different line types made of hierarchically and complementary realistic line routes, including direct routes between node pairs with the greatest demand that are currently not serviced by the rail system (referred to as A-type routes); routes linking primary transit nodes, which may include rail stations, and links identified as having the highest passenger volume (B-type routes); and routes belonging to an existing bus network (C-type routes).

In order to build such a set, a Heuristic Route Generation Algorithm (HRGA) is used [10]. It generates a large and rational set of feasible routes by applying different design criteria (e.g., efficiency and effectiveness from the point of view of both users and operators) and practical knowledge regarding the spatial arrangement of the route (for instance, the directness of the service, the length of the route, the duplication of routes, and similar factors). Then, Phase 2 applies an optimisation algorithm to find the optimal subset of routes and their frequencies within the pool of the feasibility set of routes delineated in Phase 1. Such an algorithm can be here implemented by using both the Genetic Algorithm (GA) and Particle Swarm Optimisation (PSO) approaches. It is worth stressing that Phase 2 is the same both in primary and in secondary BTND. As written, they differ in the input data used to feed the procedures: other than the transit demands, Phase 2 of the two parts differs in terms of the set of feasible routes to which the optimization procedure is applied to select the lines to be part of the design network. The whole methodology is based on the following optimisation problem, which consists of minimising all resources and costs of a public transport network in a context of rigid (constant) demand subject to a set of technical constraints.

Let:

Z be the objective function;

$\underline{r}$ be the vector of routes;

$\underline{r}^*$ be the vector of optimal routes;

$\varphi$, $\varphi^*$ be, respectively, the vector of lines' frequencies and their optimal ones;

$\overline{f}$ be the segment flows vector on the transit network;

$\underline{\Lambda}$ be the matrix of link-hyperpath traversing probabilities;

$\underline{Q}$ be the hyperpath choice probability matrix;

$\underline{c}$ be the link cost vector;

$\underline{x}^{NA}$ be the vector of hyperpath nonadditive costs;

$\underline{d}$ be the transit demand vector;

$f_{hk,l}$ be the ridership on segment (hk,l) of line l;

$fc_{max}$ be the maximum load factor;

$VC_l$ be the vehicle capacity on line l;

$\varphi_l$ be the frequency of line l with $\varphi_{min}$ and $\varphi_{max}$ representing its minimum and maximum values;

$I_a, I_l, I_{w,l}$ and $I_n$ be the sets of, respectively, the links of the network, the transit lines l, the segments of line l (*hk,l*) and nodes of the transit network;

$fw_{hk,l}$ be the boardings on segment (*hk,l*) of line l;

$tt_{hk,l}, tw_{hk,l}$ be, respectively, the travel and the waiting times for a segment of line l (*hk,l*);

$nt_n$ be the transfers at node n;

$TP_t$ be the time penalty related to a transfer;

$fa_{hk}$ be the pedestrians' flow at the link (*hk*);

$ta_{hk}$ be the access time at the link (*hk*);

$TP_u$ be the time penalty associated with an unsatisfied transit user;

$D_u$ be the unsatisfied transit demand;

$C_{km}$, $C_h$ be the factor of unit cost depending, respectively, on the total bus travel distance (vehicle operating cost) and on the bus service's total time (cost of travelling personnel);

$C_u$ be the users' average monetary value of time;

$\gamma_1$, $\gamma_2$ and $\gamma_3$ be the set of weights reflecting the relative importance given by the decision maker to each of the objective function terms $z_1$, $z_2$ and $z_3$.

Then, the optimization problem can be formally defined as

$$\left(\underline{r}^{*}, \underline{\varphi}^{*}\right) = \arg\min Z\left(\underline{r}, \underline{\varphi}, \underline{f}\right) \tag{1}$$

Subject to the following feasibility constraints, which are:

The demand-supply consistency (i.e., user equilibrium on the transit network):

$$\underline{f} = \underline{\Delta}\, \underline{Q}\left[\underline{\Delta}^{T}\underline{c}\left(\underline{r}, \underline{\varphi}\right) - \underline{x}^{NA}\left(\underline{r}, \underline{\varphi}\right)\right]\underline{d} \tag{2}$$

The technical constraints on the bus's capacity:

$$\frac{f_{hk,l}}{\varphi_l \cdot VC_l} \leq fc_{max} \tag{3}$$

And the technical constraints on bus services (i.e., both minimum and maximal values for bus frequency):

$$\varphi_{min} \leq \varphi_l \leq \varphi_{max} \tag{4}$$

Equation (1) defines the objective function Z, which can be specified as the sum of the operator's costs $z_1$ and users' costs $z_2$ plus an additional penalty related to the level of unsatisfied demand, $z_3$, that is

$$
\begin{aligned}
Z\left(\underline{r}, \underline{\varphi}, \underline{f}^{*}\right) &= z_1\left(\underline{r}, \underline{\varphi}\right) + z_2\left(\underline{r}, \underline{\varphi}, \underline{f}^{*}\right) + z_3\left(\underline{r}, \underline{\varphi}, \underline{f}^{*}\right) = \\
&= \gamma_1 \cdot \left( C_{km} \cdot \sum_{l \in I_l} L_l \varphi_l + C_h \cdot \sum_{l \in I_l} \sum_{(hk,i) \in I_{w,l}} tt_{hk,l} \varphi_l \right) + \\
&+ \gamma_2 \cdot C_u \cdot \left( \sum_{l \in I_l} \sum_{(hk,l) \in I_{w,l}} tt_{hk,l} f_{hk,l} + \sum_{l \in I_l} \sum_{(hk,l) \in I_{w,l}} tw_{hk,l} fw_{hk,l} + TP_t \cdot \sum_{n \in I_n} nt_n + \sum_{hk \in Ia} ta_{hk} fa_{hk} \right) + \\
&+ \gamma_3 \cdot C_u \cdot \left( TP_u \cdot D_u \right)
\end{aligned}
\tag{5}
$$

Equation (2) represents the demand–supply consistency constraint (assignment constraints). It consists of transit segment flows obtained by the reproduction of user choice behaviour on transit using a frequency-based hyperpath approach (see [45]). In order to

avoid violating the conditions of existence and the uniqueness of the assignment solution, transit capacity constraints are not here explicitly considered; they are included in the heuristic design procedure (Phase 1).

Equation (3) expresses the bus's capacity constraint. The line frequency cannot exceed its maximum operational value, otherwise an overload on some line sections occurs (i.e., a higher load factor than is accepted).

Equation (4) imposes reasonable technical limits on the line frequency, on the basis that other technical limits (e.g., line length) have been satisfied in the primary BTND (Phase 1). Specifically, the line frequency must not exceed the maximum operationally implementable value because it is realistically unfeasible to maintain. On the other hand, a minimum value of frequency is considered, to avoid the perception of no service availability in the case of very low frequency services (typical of regional supply services).

In order to define the optimal set of bus routes, the optimisation problem requires a road network modelled using an oriented graph G = (N, E), in which N is the nodes set and E is the links set representing the connections between nodes. A route is a sequence of adjacent nodes in G while a line is specified through a pair $\left(\underline{r}, \underline{\varphi}\right)$. The input data are the operating and users' unit costs, and the characteristics of the road network on which the public transport services will operate as well as the public transport origin–destination demand. The output data are the bus routes (including terminals) and their frequency as well as the main public transport network indicators, including total costs and flows on the public transport network.

The reader should note that because the performance of the transit system depends on its service frequencies, which should be optimized depending on the passenger flows, an iterative assignment and frequency setting procedure must be applied.

## 4. Case Study

In order to test the method, a real-world case study in the city of Rome has been carried out. The demand was taken from COVID-19 experiences, specifically, the demand for trips between the months of May 2020 and July 2020. These demand matrices are characterized by the application of different policies regarding different timings for shopping and schools' openings as well as for work-at-home policies (smart working). Specifically, the first demand matrix represents the minimum level of transit demand, characterised by a reduction of about 73% of trips with respect to the pre-COVID-19 period, while the second one concerns a decrease of about 39% of transit demand.

These matrices were estimated in previous studies to support decision-makers in defining actions for a safe restart of activities in the post-COVID-19 period in Rome [46]. They were estimated using multi-step demand models based on a system of generation, distribution and modal choice models, considering four different trip purposes and four mode alternatives. Then, these matrices per mode were updated using all traffic data available, including Floating Car Data, smartphone data and metro counts at entering gates. Moreover, in order to improve the quality of estimates, a further adjustment by means of a pivot technique was applied. The demand–supply interaction was simulated using the hyperpath choice model for public transport.

The case study proposed the running of the overall procedure with a single application of primary BTND, referring to the pre-COVID-19 level of demand (the initial solution) and a double application of secondary BTND for the above-mentioned two levels of reduced demand. It is worth mentioning that the term "reduced demand" refers to transit trips resulting from the adoption of infection control policies which imply abrupt variations in the level (usually reduced, on the whole) and distribution of transit demand.

In order to validate the procedure, two additional tests have been carried out, using primary BTND, based on the two different levels of demand. In other words, the results of the proposed procedure can be compared with the applications of the primary BTND procedure alone using the same basin of feasible routes with the other reduced demand matrices.

The resulting optimal main bus network is composed of 85 lines selected from the large set of 535 feasible routes. The tests carried out, starting from this result, are summarized as follows:

- Test 1–85, where the demand was decreased by 73% with respect to the initial level and the procedure started from the pre-selected 85 lines;
- Test 2–85, where the demand was decreased by 39% with respect to the initial level and the procedure started from the pre-selected 85 lines;
- Test 1–535, where the demand was decreased by 73% and the procedure started from the whole basin of 535 lines; and finally,
- Test 2–535, where the demand was decreased by 39% and the procedure started from the whole basin of 535 routes.

The results of such tests show the efficacy of the proposed methodology and prove its possible application on realistically sized transit networks. The application of secondary BTND has been carried out with two tests according to the two reduced transit demands, in which the optimal network was made of 40 lines in the first test and 60 lines in the second one. These values represent the best number of lines for those levels of demand derived by carrying out several GA trials varying the number of lines. These number of lines are also used for the application of primary BTND with a reduced demand.

In terms of transport analysis, the design networks obtained with the reduced demand are easily implemented, starting from the initial configuration of the supply, as a subset of the main network lines. The changes are only due to the removal of some lines, respectively, 25 and 45 with respect to the initial 85, and the variation of the frequency for part of the remaining lines. It is possible to highlight very different results when analysing the lines obtained from the application of primary BTND within the same large basin; the Test 2–535 presents only 27 of the 60 lines refined to match the initial 85-line network, while Test 1–535 has only 17 of the 40 lines that also attempt to maintain the initial 85-line network. Furthermore, the comparison between the obtained 60 and 40 lines shows that only 17 lines are overlapping. This means that such solutions are not easy to implement and that the solutions provided are without strong route overlapping. However, the proposed procedure allows us to maintain a strong relationship among the resulting bus networks, starting from a reduced set of lines generated from Phase 1 of TND.

The quality of the solution proposed by the procedure is also clearly shown by the comparison of the objective function ($Z$) value reached in the different tests carried out with different sets of feasible routes. Figure 2 shows these data for the two tests starting from the two initial configurations, highlighted in green (small basin, 85 lines) and orange (large basin, 535 lines).

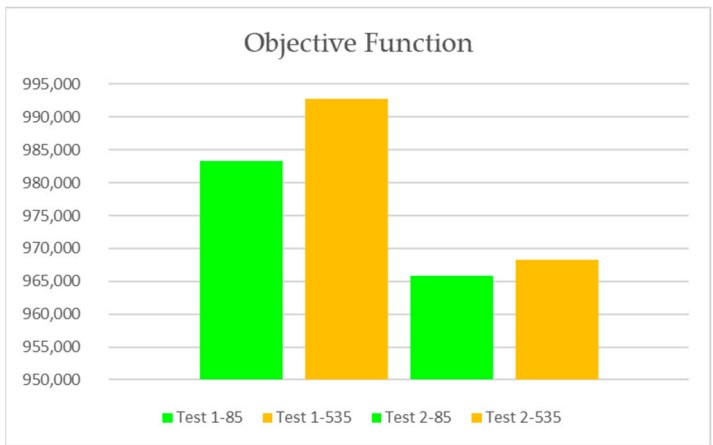

**Figure 2.** Comparison of objective functions in different tests.

As can be observed, in both cases, the solution proposed by secondary BTND results in a slightly better design than the other ones that have been provided by a very efficient solving procedure that has been tested in many numerical applications in recent years [10].

Figures 3–5 show the bus networks, respectively, consisting of 85, 40 and 60 lines, resulting from the application of the proposed method.

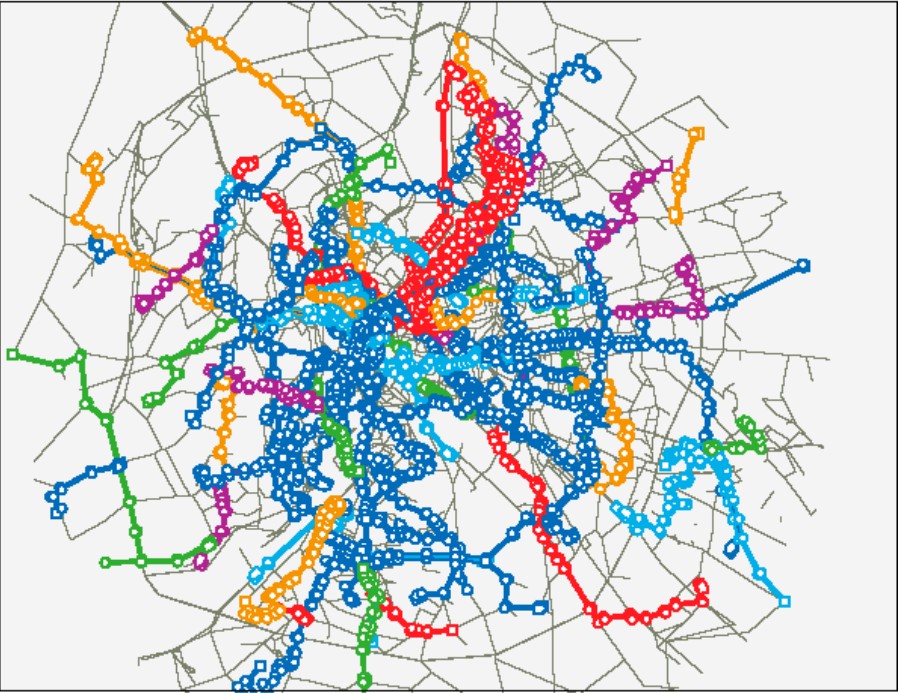

**Figure 3.** Designed bus network with 85 lines—initial solution.

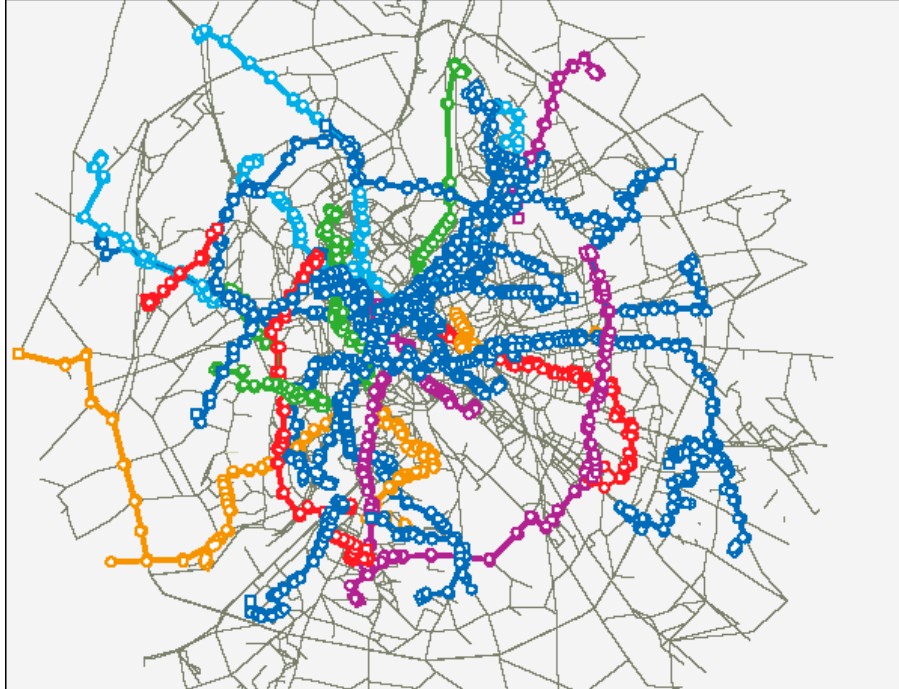

**Figure 4.** Designed bus network with 40 lines—basin of 85 lines, Test 1–85.

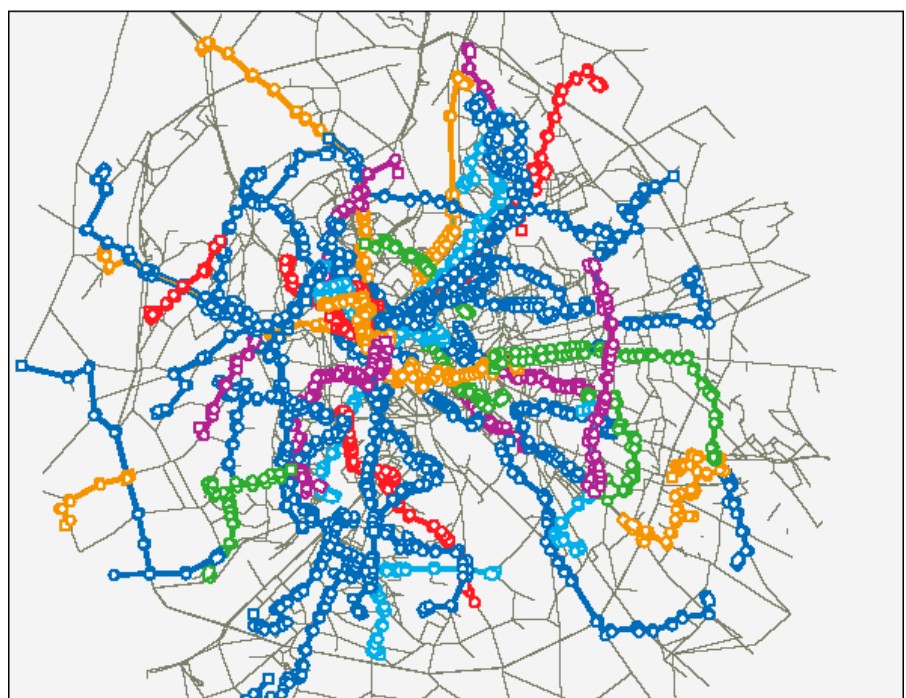

**Figure 5.** Designed bus network with 60 lines—basin of 85 lines, Test 2–85.

The following Tables 1 and 2 show the synthesis of the results from the two tests carried out with the two different initial configurations, in terms of macro-indicators about the level of supply and in terms of the level of demand not served. The threshold of this unsatisfied demand has been set at 5%. However, it is noted that actually this demand is not actually not served but rather poorly served; specifically, with a service time longer than 75 min. As it can be observed, the vehicles' km are decreasing from the 85-line network to the 40-line network.

**Table 1.** Synthesis of results—basin of 85 lines.

|  | Supply | Veh—h | Veh—km | Unsatisfied Demand [pax/h] |
|---|---|---|---|---|
| **Initial Solution** | 85 lines | 875 | 14,883 | 5817 |
| **Test 2–85** | 60 lines | 759 | 12,983 | 5938 |
| **Test 1–85** | 40 lines | 636 | 10,983 | 6759 |

**Table 2.** Synthesis of results—basin of 535 lines.

|  | Supply | Veh—h | Veh—km | Unsatisfied Demand [pax/h] |
|---|---|---|---|---|
| **Test 2–535** | 60 lines | 812 | 14,020 | 6281 |
| **Test 1–535** | 40 lines | 622 | 10,571 | 7123 |

As can be observed, the results of the procedure seem to be more efficient in terms of the supply level with respect to the solutions proposed using the large basin of 535 lines, especially for the 60-line network. These configurations are also valid from the point of view of the service provided to users because the level of not-served demand, expressed in terms of passengers per hour [pax/h] in Tables 1 and 2, is limited and very similar in all the network configurations generated.

## 5. Conclusions and Further Developments

This paper proposed a methodology for a bus transit network design able to consider demand variations from a planning point of view. It stems from the changes in our lives imposed by the COVID-19 pandemic, which is influencing our "new normal". Such an

experience has imposed a rethinking of the methodology used for the optimal design of robust transit network services that are easy to adapt to demand variations. It implies a need to manage adaptations in transit services consistently, with demand changes in both level and distribution due to variations induced by different timings for shopping and schools' openings, as well as by work-at-home policies (smart working). Therefore, this study proposed a methodology articulated in two parts: the first part, called primary BTND, sought to solve the problem for the maximum level of transit demand, aiming at giving an upper bound to the solution, while the second part, called secondary BTND, was where the network was optimized for other specific transit demands (e.g., in terms of levels, usually reduced, and distributions).

This method has been applied to a real context in the city of Rome, considering two levels of demand taken from COVID-19 experiences. These levels of demand are characterized by the application of different policies regarding different timings for shopping and schools' openings, as well as work-at-home policies (smart working). The results of such tests show the effectiveness of the proposed methodology to design robust transit networks suited to comply with large demand variations. Our promising results show that the procedure is suitable and easy to implement, in order to adapt quickly to changes in demand without having to modify line routes, and can adapt in an optimal way, even dealing with realistic-sized transit networks. This method allows to make the supply system more effective from the users' point of view and more efficient from the operators' point of view, satisfying the requests of citizens and remaining competitive with private transport, thus, the urban mobility system can become more sustainable.

Further developments will involve a sensitivity analysis of the parameters used to assess whether the optimal solutions are suitably stable when these parameters are perturbed. Moreover, this methodology can also be studied to explicitly design for peak and off-peak hours.

**Author Contributions:** Conceptualization, E.C. and M.P.; methodology, E.C., U.C., A.G., L.M. and M.P.; software, A.G. and M.P.; validation, E.C., U.C., A.G., L.M. and M.P.; formal analysis, L.M. and M.P.; investigation, U.C., A.G. and M.P.; resources, E.C., U.C., A.G., L.M. and M.P.; data curation, A.G., L.M. and M.P.; writing—original draft preparation, E.C., U.C., A.G., L.M. and M.P.; supervision, E.C. and M.P. All authors have read and agreed to the published version of the manuscript.

**Funding:** This research received no external funding.

**Institutional Review Board Statement:** Not applicable.

**Informed Consent Statement:** Not applicable.

**Data Availability Statement:** Data are contained within the article.

**Conflicts of Interest:** The authors declare no conflicts of interest.

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
