# Peer review of "A Bus Network Design Model under Demand Variation: A Case Study of the Management of Rome’s Bus Network"

_sustainability, doi:10.3390/su16020803_

Round 1

Reviewer 1 Report

Comments and Suggestions for Authors
  1. The image is blurry, please use a higher resolution image. Such as Figure 1 and 3,4,5.
  2. Mathematical characters in the manuscript should use a formula editor to ensure beautiful typography. The most important thing is that the same symbol should have the same representation in the body and in the formula, for example, line 257 uses "z1, z2, z3", but the formula after line 258 uses "$z_1$, $z_2$, $z_3$".
  3. The author uses more than one page (page5&6) to explain symbols, please organize it into a table and reduce unnecessary parts.
  4. Please use Eq or Eqs as abbreviations for the word Equation, instead of Eqn.
  5. In case stduy, the author only presents his own results, but it lacks comparison with the existing methods in literature review, which cannot highlight the contribution and innovation of the manuscript.
  6. Please pay more attention to the latest research, all the references listed are before 2020. Please refer to: https://doi.org/10.1016/j.compenvurbsys.2022.101776  , and, DOI: 10.1109 / TITS. 2022.3145655

Comments on the Quality of English Language

Moderate editing of English language required

Reviewer 2 Report

Comments and Suggestions for Authors

at 338 please explain w.r.t.

the main idea of this paper is to provide a procedure of
transit network design that could consider the large demand variations

the approach based on two phases is not new but the use of GA in doing that it is so the study represent a contribution to the current field

perhaps it will be a good idea for authors to have a baseline method  needed in order to place the efficiency of the study among another's. Or at least to present comparative a mathematical analysis.

the conclusions are in accordance with the proposed model and the presented experimental results

the references are good but the lack of high impact journals (such as e.g. Elsevier) limit the authors state of the art value

Reviewer 3 Report

Comments and Suggestions for Authors

The authors discussed a novel hierarchical design in the transit networks, which aims to address the challenges imposed by the COVID-19 pandemic and improve an overall optimization score that consolidates transit network operational cost, users' cost, and users' unsatisfied demand. Please see my comments below.

1. In the formula (5), the authors include a series of constant parameters, e.g., Cu, the average monetary value of time for the users. However, it seems unclear to me how authors decide their values, and whether or not these decisions are based upon any existing literature. The authors should also conduct sensitivity analysis to validate whether the optimal solutions are sufficiently stable when those parameters are perturbed.

2. The objective function Z can be viewed as addition of 3 sub-optimization problems: 1) minimizing the operational cost, 2) minimizing the users' cost, and 3) minimizing users' unsatisfied demand. This is essentially a multiple objective task. The authors should briefly explain why they choose the linear scalarization approach but not take other multi-objective optimization approaches.

3. In Table 1, it seems that both tests 1 and 2 yield higher unsatisfied demand than the initial solution. I was wondering if this is caused by insufficient penalty related to unsatisfied demand in the objective function. The authors should explain why we observe this and what level of increase in unsatisfied demand is acceptable.

4. In figure 2, why exclude the objective function of the initial solution?

5. What is the optimization algorithm in finding the optimal solution? Is the optimal solution deterministic or stochastic? If non-deterministic, the authors should provide a confidence interval on the metrics they have presented.

Reviewer 4 Report

Comments and Suggestions for Authors

The authors of this article have presented a methodology aimed at designing bus public transportation networks that can dynamically adjust to variations in the level and distribution of demand. The objective is to assist service providers in optimizing bus network capacity through a comprehensive systemwide analysis. The results obtained demonstrate the effectiveness of the proposed methodology in creating robust public transportation networks capable of adapting to significant variations in demand.

In my opinion, the topic is engaging, the article is well-structured, effectively presented, and persuasive. I recommend its publication.

Author Response

Thank you for your comment.

Reviewer 5 Report

Comments and Suggestions for Authors

After reading the manuscript the following improvements could be done:

-          Please include the equation number in the same line where the equation is written.

-          In page 7 subscripts should be used to refer to z1, z2, and z3.

-          A discussion of the test results was given in the manuscript. However, a discussion about how the data was gathered should be given. This will give the reader a better understanding of the test results.

-          Please show the summary of the results and graphs before discussing the outcome the analysis.

-          The reference figure 2 properly in the text.

-           Please explain in the figure caption the units pax/h used in Tables 1 and 2.

-          Please include in the conclusions sections future progress that could be done with the methodology discussed in the manuscript.   

Reviewer 6 Report

Comments and Suggestions for Authors

1.  Review the relevance and appropriateness of the case study (Rome bus network during the COVID-19 pandemic).  Does it effectively demonstrate the application of the methodology?

2.  Are the findings presented in a clear, logical manner?

3.  Assess the paper's contribution to the field of urban transit network management.  Does it offer new insights or solutions?

4.  Recommend ways to strengthen the argument about the paper’s significance and relevance, especially in the context of changing urban mobility patterns due to global crises.

5.  Suggest any additional literature that the authors might have overlooked, which could provide further context or support for their methodology and findings.

6.  Suggest improvements or additional graphical representations that could enhance the reader's understanding.

Comments on the Quality of English Language

1.  Review the relevance and appropriateness of the case study (Rome bus network during the COVID-19 pandemic).  Does it effectively demonstrate the application of the methodology?

2.  Are the findings presented in a clear, logical manner?

3.  Assess the paper's contribution to the field of urban transit network management.  Does it offer new insights or solutions?

4.  Recommend ways to strengthen the argument about the paper’s significance and relevance, especially in the context of changing urban mobility patterns due to global crises.

5.  Suggest any additional literature that the authors might have overlooked, which could provide further context or support for their methodology and findings.

6.  Suggest improvements or additional graphical representations that could enhance the reader's understanding.

Round 2

Reviewer 1 Report

Comments and Suggestions for Authors

all issue are revised and there are no new more comments.